# FIX YOUR CLASSIFIER: THE MARGINAL VALUE OF TRAINING THE LAST WEIGHT LAYER

**Elad Hoffer, Itay Hubara, Daniel Soudry**
Department of Electrical Engineering
Technion
Haifa, 320003, Israel
`elad.hoffer, itay.hubara, daniel.soudry@gmail.com`

## ABSTRACT

Neural networks are commonly used as models for classification for a wide variety of tasks. Typically, a learned affine transformation is placed at the end of such models, yielding a per-class value used for classification. This classifier can have a vast number of parameters, which grows linearly with the number of possible classes, thus requiring increasingly more resources.

In this work we argue that this classifier can be fixed, up to a global scale constant, with little or no loss of accuracy for most tasks, allowing memory and computational benefits. Moreover, we show that by initializing the classifier with a Hadamard matrix we can speed up inference as well. We discuss the implications for current understanding of neural network models.

## 1 INTRODUCTION

Deep neural network have become a widely used model for machine learning, achieving state-of-the-art results on many tasks. The most common task these models are used for is to perform classification, as in the case of convolutional neural networks (CNNs) used to classify images to a semantic category. CNN models are currently considered the standard for visual tasks, allowing far better accuracy than preceding approaches (Krizhevsky et al., 2012; He et al., 2016; Szegedy et al., 2015).

Training NN models and using them for inference requires large amounts of memory and computational resources, thus, extensive amount of research has been done lately to reduce the size of networks. Han et al. (2015) used weight sharing and specification, Micikevicius et al. (2017) used mixed precision to reduce the size of the neural networks by half. Tai et al. (2015) and Jaderberg et al. (2014) used low rank approximations to speed up NNs.

Hubara et al. (2016b), Li et al. (2016) and Zhou et al. (2016), used a more aggressive approach, in which weights, activations and gradients were quantized to further reduce computation during training. Although aggressive quantization benefits from smaller model size, the extreme compression rate comes with a loss of accuracy.

Past work noted the fact that predefined (Park & Sandberg, 1991) and random (Huang et al., 2006) projections can be used together with a learned affine transformation to achieve competitive results on several tasks. In this study suggest the reversed proposal - that common NN models used can learn useful representation even without modifying the final output layer, which often holds a large number of parameters that grows linearly with number of classes.

### 1.1 CLASSIFIERS IN CONVOLUTIONAL NEURAL NETWORKS

Convolutional neural networks (CNNs) are commonly used to solve a variety of spatial and temporal tasks. CNNs are usually composed of a stack of convolutional parameterized layers, spatial pooling layers and fully connected layers, separated by non-linear activation functions. Earlier architectures of CNNs (LeCun et al., 1998; Krizhevsky et al., 2012) used a set of fully-connected layers at later stage of the network, presumably to allow classification based on global features of an image. The

final classifier can also be replaced with a convolutional layer with output feature maps matching the number of classes, as demonstrated by Springenberg et al. (2014).

Despite the enormous number of trainable parameters these layers added to the model, they are known to have a rather marginal impact on the final performance of the network (Zeiler & Fergus, 2014) and are easily compressed and reduced after a model was trained by simple means such as matrix decomposition and sparsification (Han et al., 2015). Further more, modern architecture choices are characterized with the removal of most of the fully connected layers (Lin et al., 2013; Szegedy et al., 2015; He et al., 2016), which was found to lead to better generalization and overall accuracy, together with a huge decrease in the number of trainable parameters.

Additionally, numerous works showed that CNNs can be trained in a metric learning regime (Bromley et al., 1994; Schroff et al., 2015; Hoffer & Ailon, 2015), where no explicit classification layer was introduced and the objective regarded only distance measures between intermediate representations. Hardt & Ma (2017) suggested an all-convolutional network variant, where they kept the original initialization of the classification layer fixed with no negative impact on performance on the Cifar10 dataset. All of these properties provide evidence that fully-connected layers are in fact redundant and play a small role in learning and generalization.

Despite the apparent minor role they play, fully-connected layers are still commonly used as classification layers, transforming from the dimension of network features $N$ to the number of required class categories $C$. Therefore, each classification model must hold $N \cdot C$ number of trainable parameters that grows in a linear manner with the number of classes. This property still holds when the fully-connected layer is replaced with a convolutional classifier as shown by Springenberg et al. (2014).

In this work we claim that for common use-cases of convolutional network, the parameters used for the final classification transform are completely redundant, and can be replaced with a predetermined linear transform. As we will show for the first time, this property holds even in large-scale models and classification tasks, such as recent architectures trained on the ImageNet benchmark (Deng et al., 2009).

The use of a fixed transform can, in many cases, allow a huge decrease in model parameters, and a possible computational benefit. We suggest that existing models can, with no other modification, devoid their classifier weights, which can help the deployment of those models in devices with low computation ability and smaller memory capacity. Moreover, as we keep the classifier fixed, less parameters need to be updated, reducing the communication cost for models deployed in distributed systems. The use of a fixed transform which does not depend on the number classes can allow models to scale to a large number of possible outputs, without a linear cost in the number of parameters. We also suggest that these finding might shed light on the importance of the preceding non-linear layers to learning and generalization.

## 2  USING A FIXED CLASSIFIER

### 2.1  FULLY-CONNECTED CLASSIFIERS

We focus our attention on the final representation obtained by the network (the last hidden layer), before the classifier. We denote these representation as $x = F(z; \theta)$ where $F$ is assumed to be a deep neural network with input $z$ and parameters $\theta$, e.g., a convolutional network, trained by back-propagation.

In common NN models, this representation is followed by an additional affine transformation

$$y = W^T x + b$$

where $W$ and $b$ are also trained by back-propagation.

For input $x$ of $N$ length, and $C$ different possible outputs, $W$ is required to be a matrix of $N \times C$. Training is done using cross-entropy loss, by feeding the network outputs through a softmax activation

$$v_i = \frac{e^{y_i}}{\sum_j^C e^{y_j}}, \; i \in \{1, \ldots, C\}$$

and reducing the expected negative log likelihood with respect to ground-truth target $t \in \{1, \dots, C\}$, by minimizing

$$\mathcal{L}(x, t) = -\log v_t = -w_t \cdot x - b_t + \log \left( \sum_{j}^{C} e^{w_j \cdot x + b_j} \right)$$

where $w_i$ is the $i$-th column of $W$.

## 2.2 Choosing the projection matrix

To evaluate our conjecture regarding the importance of the final classification transformation, we replaced the trainable parameter matrix $W$ with a fixed orthonormal projection $Q \in \mathbb{R}^{N \times C}$, such that $\forall i \neq j : q_i \cdot q_j = 0$ and $\|q_i\|_2 = 1$, where $q_i$ is the $i$th column of $Q$. This can be ensured by a simple random sampling and singular-value decomposition

As the rows of classifier weight matrix are fixed with an equally valued $L_2$ norm, we find it beneficial to also restrict the representation of $x$ by normalizing it to reside on the $n$-dimensional sphere

$$\hat{x} = \frac{x}{\|x\|_2} \tag{1}$$

This allows faster training and convergence, as the network does not need to account for changes in the scale of its weights.

We now face the problem that $q_i \cdot \hat{x}$ is bounded between $-1$ and $1$. This causes convergence issues, as the softmax function is scale sensitive, and the network is affected by the inability to re-scale its input. This is similar to the phenomenon described by Vaswani et al. (2017) with respect to softmax function used for attention mechanisms. In the same spirit, we can amend this issue with a fixed scale $T$ applied to softmax inputs $f(y) = \text{softmax}(\frac{1}{T} y)$, also known as a softmax temperature. However, this introduces an additional hyper-parameter which may differ between networks and datasets. Instead, we suggest to introduce a single scalar parameter $\alpha$ to learn the softmax scale, effectively functioning as an inverse of the softmax temperature $\frac{1}{T}$.

Using normalized weights and an additional scale coefficient is similar in spirit to weight-normalization (Salimans & Kingma, 2016), with the difference that we use a single scale for all entries in the weight matrix, in contrast to a scale for each row that Salimans & Kingma (2016) uses.

We keep the additional vector of bias parameters $b \in \mathbb{R}^C$, and train using the same negative-log-likelihood criterion. More explicitly, our classifier output is now

$$v_i = \frac{e^{\alpha q_i \cdot \hat{x} + b_i}}{\sum_{j}^{C} e^{\alpha q_j \cdot \hat{x} + b_j}}, \ i \in \{1, \dots, C\}$$

and we minimize the loss:

$$\mathcal{L}(x, t) = -\alpha q_t \cdot \frac{x}{\|x\|_2} + b_t + \log \left( \sum_{i=1}^{C} \exp \left( \alpha q_i \cdot \frac{x}{\|x\|_2} + b_i \right) \right)$$

where we recall $x$ is the final representation obtained by the network for a specific sample, and $t \in \{1, \dots, C\}$ is the ground-truth label for that sample.

Observing the behavior of the $\alpha$ parameter over time revealed a logarithmic growth depicted in graph 1. Interestingly, this is the same behavior exhibited by the norm of a learned classifier, first described by Hoffer et al. (2017) and linked to the generalization of the network. This was recently explained by the under-review work of Soudry et al. (2018) as convergence to a max margin classifier. We suggest that using a single parameter will enable a simpler examination and possible further exploration of this phenomenon and its implications.

We note that as $-1 \leq q_i \cdot \hat{x} \leq 1$, we also found it possible to train the network with a simple cosine angle loss:

$$\mathcal{L}(\hat{x}, t) = \begin{cases} q_i \cdot \hat{x} - 1, & \text{if } i = t, \\ q_i \cdot \hat{x} + 1, & \text{otherwise.} \end{cases}$$

allowing to discard the softmax function and its scale altogether, but resulting in a slight decrease in final validation accuracy compared to original models.

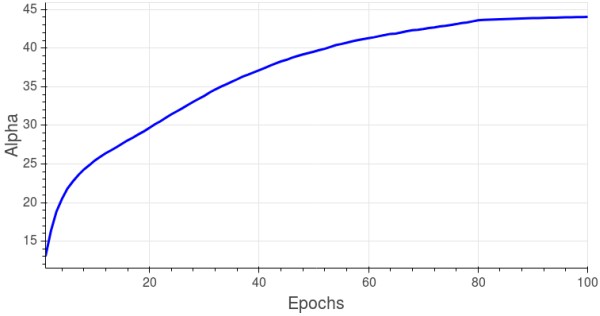

Figure 1: The softmax scale coefficient $\alpha$ was observed to follow a logarithmic growth over the course of training.

### 2.3 USING A FIXED HADMARD MATRIX

We further suggest the use of a Hadamard matrix (Hedayat et al., 1978) as the final classification transform. Hadamard matrix $H$ is an $n \times n$ matrix, where all of its entries are either $+1$ or $-1$. Further more, $H$ is orthogonal, such that $HH^T = nI_n$ where $I_n$ is the identity matrix.

We can use a truncated Hadamard matrix $\hat{H} \in \{-1, 1\}^{C \times N}$ where all $C$ rows are orthogonal as our final classification layer such that

$$y = \hat{H}\hat{x} + b$$

This usage allows two main benefits:

- A deterministic, low-memory and easily generated matrix that can be used to classify.

- Removal of the need to perform a full matrix-matrix multiplication - as multiplying by a Hadamard matrix can be done by simple sign manipulation and addition.

We note that $n$ must be a multiple of $4$, but it can be easily truncated to fit normally defined networks.

We also note the similarity of using a Hadamard matrix as a final classifier to methods of weight binarization such as the one suggested by Courbariaux et al. (2015). As the classifier weights are fixed to need only 1-bit precision, it is now possible to focus our attention on the features preceding it.

## 3 EXPERIMENTAL RESULTS

Table 1: Validation accuracy results on learned vs. fixed classifier

| Network | Dataset | Learned | Fixed | # Params | % Fixed params |
|---|---|---|---|---|---|
| Resnet56 (He et al., 2016) | Cifar10 | 93.03% | 93.14% | 855,770 | 0.07% |
| DenseNet(k=12)(Huang et al., 2017) | Cifar100 | 77.73% | 77.67% | 800,032 | 4.2% |
| Resnet50 (He et al., 2016) | ImageNet | 75.3% | 75.3% | 25,557,032 | 8.01% |
| DenseNet169(Huang et al., 2017) | ImageNet | 76.2% | 76% | 14,149,480 | 11.76% |
| ShuffleNet(Zhang et al., 2017b) | ImageNet | 65.9% | 65.4% | 1,826,555 | 52.56% |

### 3.1 CIFAR10/100

We used the well known Cifar10 and Cifar100 datasets by Krizhevsky (2009) as an initial test-bed to explore the idea of a fixed classifier. Cifar10 is an image classification benchmark dataset containing $50,000$ training images and $10,000$ test images. The images are in color and contain $32 \times 32$ pixels. There are 10 possible classes of various animals and vehicles. Cifar100 holds the same number of images of same size, but contains 100 different classes.

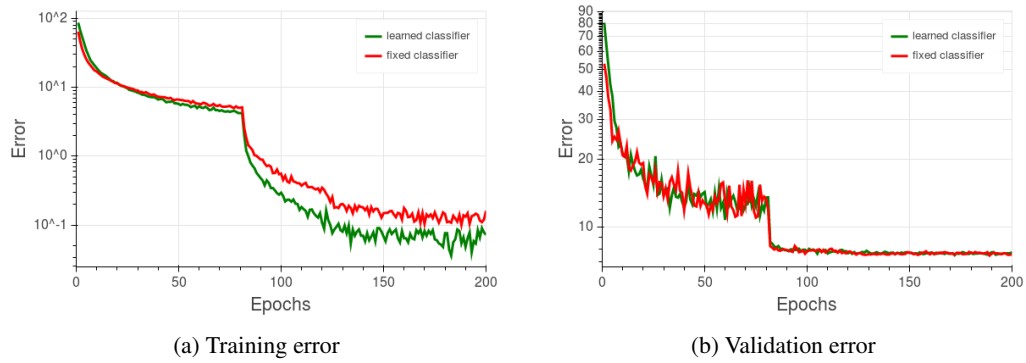

(a) Training error

(b) Validation error

Figure 2: Comparing training and validation error of fixed and learned classifier (ResNet56, Cifar10)

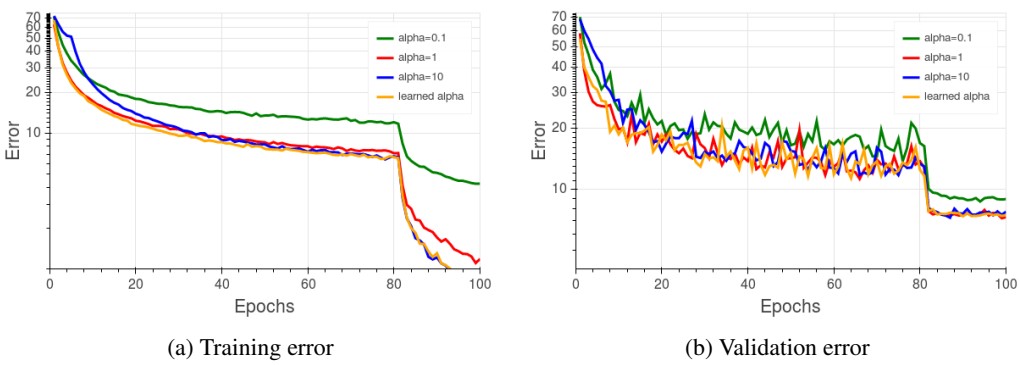

(a) Training error

(b) Validation error

Figure 3: Comparing fixed vs. trained variable scale $\alpha$ (ResNet56, Cifar10)

We trained a residual network of He et al. (2016) on the Cifar10 dataset. We used a network of depth 56 and the same hyper-parameters used in the original work. We compared two variants: the original model with a learned classifier, and our version, where a fixed transformation is used. The results shown in figure 2 demonstrate that although the training error is considerably lower for the network with learned classifier, both models achieve the same classification accuracy on the validation set. Our conjecture is that with our new fixed parameterization, the network can no longer increase the norm of a given sample's representation - thus learning its label requires more effort. As this may happen for specific seen samples - it affects only training error.

We also compared using a fixed scale variable $\alpha$ at different values vs. a learned parameter. Results for $\alpha = \{0.1, 1, 10\}$ are depicted in figure 3 for both training and validation error. As can be seen, similar validation accuracy can be obtained using a fixed scale value (in this case $\alpha = 1$ or $10$ will suffice) at the expense of another hyper-parameter to seek. In all our experiments we opted to train this parameter instead. In all experiments the $\alpha$ scale parameter was regularized with the same weight decay coefficient used on original classifier.

We then followed to train a model on the Cifar100 dataset. We used the DenseNet-BC model of Huang et al. (2017) with depth of 100 layers and $k = 12$. We continued to train according to the original regime and setting described for this network and dataset. Naturally, the higher number of classes caused the number of parameters to grow and encompass about $4\%$ of the whole model. Validation accuracy for the fixed-classifier model remained equally good as the original model, and we continued to observe the same training curve.

## 3.2 IMAGENET

In order to validate our results on a more challenging dataset, we used the Imagenet dataset introduced by Deng et al. (2009). The Imagenet dataset spans over 1000 visual classes, and over 1.2 million samples. CNNs used to classify Imagenet such as Krizhevsky et al. (2012), He et al. (2016),

Szegedy et al. (2016) usually have a hidden representation leading to the final classifier of at least 1024 dimensions. This architectural choice, together with the large number of classes, causes the size of classifier to exceed millions of parameters and taking a sizable share from the entire model size.

We evaluated our fixed classifier method on Imagenet using Resnet50 by He et al. (2016) with the same training regime and hyper-parameters. By using a fixed classifier, approximately 2-million parameters were removed from the model, accounting for about 8% of the model parameters. Following the same procedure, we trained a Densenet169 model (Huang et al., 2017) for which a fixed classifier reduced about 12% of the parameters. Similarly to results on Cifar10 dataset, we observed the same convergence speed and approximately the same final accuracy on both the validation and training sets.

Furthermore, we were interested in evaluating more challenging models where the classifier parameters constitutes the majority amount. For this reason we chose the Shufflenet architecture (Zhang et al., 2017b), which was designed to be used in low memory and limited computing platforms. The Shufflenet network contains about 1.8 million parameters, out of which 0.96 million are part of the final classifier. Fixing the classifier resulted with a model with only 0.86 million parameters. This model was trained and found, again, to converge to similar validation accuracy as the original.

Interestingly, this method allowed Imagenet training in an under-specified regime, where there are more training samples than number of parameters. This is an unconventional regime for modern deep networks, which are usually over-specified to have many more parameters than training samples (Zhang et al., 2017a). Moreover, many recent theoretical results related to neural network training (Soudry & Hoffer, 2017; Xie et al., 2016; Safran & Shamir, 2016; Soltanolkotabi et al., 2017; Soudry & Carmon, 2016) and even generalization (Gunasekar et al., 2017; Advani & Saxe, 2017; Wilson et al., 2017) usually assume over-specification.

Table 1 summarizes our fixed-classifier results on convolutional networks, comparing to originally reported results. We offer our drop-in replacement for learned classifier that can be used to train models with fixed classifiers and replicate our results[1].

## 3.3 LANGUAGE MODELING

As language modeling requires classification of all possible tokens available in the task vocabulary, we were interested to see if a fixed classifier can be used, possible saving a very large number of trainable parameters (vocabulary size can have tens or even hundreds of thousands of different words). Recent works have already found empirically that using the same weights for both word embedding and classifier can yield equal or better results than using a separate pair of weights (Inan et al., 2016; Press & Wolf, 2017; Vaswani et al., 2017). This is compliant with our findings that the linear classifier is largely redundant. To examine further reduction in the number of parameters, we removed both classifier and embedding weights and replaced them with a fixed transform.

We trained a language model on the WikiText2 dataset described in Merity et al. (2016), using the same setting in Merity et al. (2017). We used a recurrent model with 2-layers of LSTM (Hochreiter & Schmidhuber, 1997) and embedding + hidden size of 512. As the vocabulary of WikiText2 holds about $33K$ different words, the expected number of parameters in embedding and classifier is about 34-million. This number makes for about $89\%$ from the $38M$ parameters used for the whole model.

We found that using a random orthogonal transform yielded poor results compared to learned embedding. We suspect that, in oppose to image classification benchmarks, the embedding layer in language models holds information of the words similarities and relations, thus requiring a fine initialization. To test our intuition, we opted to use pre-trained embeddings using word2vec algorithm by Mikolov et al. (2013) or PMI factorization as suggested by Levy & Goldberg (2014). We find that using fixed word2vec embeddings, we achieve much better results. Specifically, we use $89\%$ less parameters than the fully learned model, and obtain only somewhat worse perplexity.

We argue that this implies a required structure in word embedding that stems from semantic relatedness between words and the natural imbalance between classes. However, we suggest that with

---

[1]Code is available at `https://github.com/eladhoffer/fix_your_classifier`

a much more cost effective ways to train word embeddings (e.g., Mikolov et al. (2013)), we can narrow the gap and avoid their cost when training bigger models.

Table 2: Validation perplexity results

| Network | Dataset | Learned | Fixed | # Params | % Fixed params |
|---|---|---|---|---|---|
| 2-layer LSTM (h=512) | WikiText-2 | 74.1 | 81.2 | 38,312,446 | 88.94% |

## 4 DISCUSSION

### 4.1 IMPLICATIONS TO FUTURE DNN MODELS AND USE CASES

In the last couple of years a we observe a rapid growth in the number of classes benchmark datasets contain, for example: Cifar100 (Krizhevsky, 2009), ImageNet1K, ImageNet22k (Deng et al., 2009) and language modeling (Merity et al., 2016). Therefore the computational demands of the final classifier will increase as well and should be considered no less than the architecture chosen. We use the work by Sun et al. (2017) as our use case, which introduced JFT-300M - an internal Google dataset with over 18K different classes. Using a Resnet50 (He et al., 2016), with a 2048 sized representation, this led to a model with over 36M parameters. This means that over $60\%$ of the model parameters reside in the final classification layer.

Sun et al. (2017) further describes the difficulty in distributing this amount of parameters between the training servers, and the need to split them between 50 sub-layers. We also note the fact that the training procedure needs to account for synchronization after each parameter update - which must incur a non-trivial overhead.

Our work can help considerably in this kind of scenario - where using a fixed classifier removes the need to do any gradient synchronization for the final layer. Furthermore, using a Hadamard matrix, we can remove the need to save the transformation altogether, and make it more efficient, allowing considerable memory and computational savings.

### 4.2 POSSIBLE CAVEATS

We argue that our method works due to the ability of preceding layers in the network to learn separable representations that are easily classified even when the classifier itself is fixed. This property can be affected when the ratio between learned features and number of classes is small – that is, when $C > N$. We've been experimenting with such cases, for example Imagenet classification ($C = 1000$) using mobilenet-0.5 (Howard et al., 2017) where $N = 512$, or reduced version of ResNet (He et al., 2016) where $N = 256$. In both scenarios, our method converged similarly to a fully learned classifier reaching the same final validation accuracy. This is strengthening our finding, showing that even in cases in which $C > N$, fixed classifier can provide equally good results.

Another possible issue may appear when the possible classes are highly correlated. As a fixed orthogonal classifier does not account for this kind of correlation, it may prove hard for the network to learn in this case. This may suggest another reason for the difficulties we experienced in training a language model using an orthogonal fixed classifier, as word classes tend to have highly correlated instances.

### 4.3 FUTURE WORK

Understanding that linear classifiers used in NN models are largely redundant allows us to consider new approaches in training and understanding these models.

Recent works (Neyshabur et al., 2017; Bartlett et al., 2017) suggested a connection between generalization capabilities of models and various norm-related quantities of their weights. Such results might be potentially simplified in our model, since we have a single scalar variable (i.e., scale), which seems to be the only relevant parameter in the model (since we normalize the last hidden layer, and fix the last weight layer).

The use of fixed classifiers might be further simplified in Binarized Neural Networks (Hubara et al., 2016a), where the activations and weights are restricted to $\pm 1$ during propagations. In this case the norm of the last hidden layer is constant for all samples (equal to the square root of the hidden layer width). This constant can be absorbed into the scale constant $\alpha$, and there is no need in a per-sample normalization as in eq. 1.

We also plan to further explore more efficient ways to learn word embedding, where similar redundancy in classifier weights may suggest simpler forms of token representations - such as low-rank or sparse versions, allowing similar benefits to the fixed transformations we suggested.

## 5    CONCLUSION

In this work we suggested removing the parameters from the classification layer used in deep neural networks. We showed empirical results suggesting that keeping the classifier fixed cause little or no decline in classification performance for common balanced datasets such as Cifar and Imagenet, while allowing a noticeable reduction in trainable parameters. We argue that fixing the last layer can reduce the computational complexity for training as well as the communication cost in distributed learning. Furthermore, using a Hadamard matrix as classifier might lead to some computational benefits when properly implemented, and save memory otherwise spent on large amount of transformation coefficients. As datasets tend to become more complex by time (e.g., Cifar100, ImageNet1K, ImageNet22k, JFT-300M, and language modeling) we believe that resource hungry affine transformation should remain fixed during training, at least partially.

We also found that new efficient methods to create pre-defined word embeddings should be explored, as they require huge amount of parameters that can possibly be avoided when learning a new task. Based on these findings, we recommend future research to focus on representations learned by the non-linear part of neural networks - up to the final classifier, as it seems to be highly redundant.

### ACKNOWLEDGMENTS

The research leading to these results has received funding from the Taub Foundation, and the European Research Council under European Unions Horizon 2020 Program, ERC Grant agreement no. 682203 SpeedInfTradeoff.

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
