# OpenReview forum: "Fix your classifier: the marginal value of training the last weight layer"
_ICLR.cc/2018/Conference — Accept (Poster)_

### Official Review · AnonReviewer3 · 2017-11-27
**Interesting idea**

**Rating:** 6
**Confidence:** 5

**Review:**

The paper proposes to use a fixed weight matrix to replace the final linear projection in a deep neural network.
This fixed classifier is combined with a global scaling and per output shift that are learned.
The authors claim that this can be used as a drop in replacement for standard architectures and does not result in reduced performance.
The key advantage is that it generates a reduction in parameters (e.g. for resent 50 8% of parameters are eliminated).

The idea is extremely simple and I like it conceptually.
Currently it looks like my reimplementation on resent 50 is working.
I do lose a about 1% in accuracy compared to my baseline learned projection implementation.
Is the scale and bias regularized?

I have assigned a score of 6 now.  but I will wait for my final rating when I get the actual results.
Overall the evaluation is seems reasonably thorough many tasks were presented and the model was applied to different architectures.

I also think the manuscript could benefit from the following experiments:
- how does the chosen projection matrix affect performance.
- is the scale needed
I assume the authors did these experiments when they developed the method but it is unclear how important these choices are.
Including these experiments would make it a more scientific contribution.

The amount of computation saved seems rather limited? Especially since the gradient of the scale parameter has to go through the weight vector?
Therefore my assumption is that only the application of the gradients save a limited amount of time and memory?
At least in my experiments reproducing these results, the computational benefit is not there/obvious.

While I like the idea, the way the manuscript is written is a bit strange at times.
The introduction appears to be there to be because you need a introduction, not to explain the background.
For this reason some of the cited work seems a bit out of place.
Especially the universal approximation and data memorization references.
What I find interesting is that this work is the complement of the reservoir computing/extreme learning machines approach.
There the final output layer is trained but the network itself uses random weights.

It would be nice if Fig 2 had a better caption. Which dataset, model, ….
Is there an intuition why the training error remains higher but the validation error is identical? This is difficult to get my head round.
Also, it would be nice if an analysis was provided where the computational cost of not doing the gradient update was computed.

---

> ### Author Response · Authors · 2017-12-23
> **answer**
>
> We thank the reviewer for his detailed feedback on our paper and his suggestions. We hope to answer his questions below. We also made adjustments to latest revision accordingly.
>
> 1) "Are the scale and bias regularized?" - Yes. We found that regularization can help with the final validation error in the same way it helps with common learned weights. Best results appeared when trained with weight decay for several epochs and removed later.
>
> 2) "how does the chosen projection matrix affect performance" - We found no significance change in final accuracy when using different projection matrix. We do find slight change in convergence rate when initial scale is changed.
>
> 3) "is the scale needed" - We added some experiments to show that the scale is not needed as a learned parameter, but this may help convergence.
>
> 4) "The amount of computation saved seems rather limited?" - The compute saved is for the gradient of the classifier weights (which is not needed to get the gradient for the scale). This may be limited for the cases shown, but becomes more apparent when number of classes is larger. As we noted, these gradients and weights can now be avoided in communication over several nodes in distributed setting - saving precious bandwidth. Moreover using a Hadamard matrix we can replace all multiplication operations preformed by the classifier with additions which are far more hardware friendly.
>
> 5)"Is there an intuition why the training error remains higher but the validation error is identical?" - Our conjecture is that with our new fixed parameterization, the network can no longer increase the norm of a given sample's representation - thus learning its label requires more effort. As this may happen for specific seen samples - it affects only training error.
>
> Regarding clarity and manuscript structure - we have taken the reviewer's comments into account and revised our paper accordingly.

---

> > ### Comment · AnonReviewer3 · 2018-01-12
> > **answer**
> >
> > Thank you for adding the additional experiments.
> >
> > I will not modify the score.
> > I still believe the idea is interesting, but it is unclear how large the impact actually is on performance.
> > In my experiments, I observed a small loss in accuracy but no improvement in speed.
> >
> > Currently, many papers on large batch training show close to linear scaling. Especially the FB in 1 hour approach where the gradient updates for higher layers are communicated in parallel with the gradient computation for lower layers.  So it is not clear how much of a difference not doing back-propagation would make.
> >
> > Ideally I would suggest the authors to implement a cuda kernel for the hadamard transform too show that the speed up is effectively there.

---

### Official Review · AnonReviewer2 · 2017-11-27
**.**

**Rating:** 6
**Confidence:** 3

**Review:**

This paper proposes replacing the weights of the final classifier layer in a CNN with a fixed projection matrix.  In particular a Hadamard matrix can be used, which can be represented implicitly.

I'd have liked to see some discussion of how to efficiently implement the Hadamard transform when the number of penultimate features does not match the number of classes, since the provided code does not do this.

How does this approach scale as the number of classes grows very large (as it would in language modeling, for example)?

An interesting experiment to do here would be to look this technique interacts with distillation, when used in the teacher or student network or both.   Does fixing the features make it more difficult to place dog than on boat when classifying a cat?  Do networks with fixed classifier weights make worse teachers for distillation?

---

> ### Author Response · Authors · 2017-12-23
> **answer**
>
> We thank the reviewer for his feedback and suggestions. We added an explanation as well as extended the supplementary code for the case where number of penultimate features does not match the number of classes.
> We also added to the discussion the case where C >> N. Regarding distillation - we found no apparent difference when distilling a network with fixed classifier.

---

### Official Review · AnonReviewer1 · 2017-12-01
**Good experiments, concerns about novelty and scalability**

**Rating:** 6
**Confidence:** 4

**Review:**

Revised Review:

The authors have largely addressed my concerns with the revised manuscript. I still have some doubts about the C > N setting (the new settings of C / N of 4 and 2 aren't C >> N, and the associated results aren't detailed clearly in the paper), but I think the paper warrants acceptance.

Original Review:

The paper proposes fixing the classification layers of neural networks, replacing the traditional learned affine transformation with a fixed (e.g., Hadamard) matrix. This is motivated by the observation that classification layers frequently constitute a non-trivial fraction of a network's overall parameter count, compute requirements, and memory usage, and by the observation that removal of pre-classification fully-connected layers has often been found to have minimal impact on performance. Experiments are performed on a range of datasets and network architectures, in both image classification and NLP settings.

First, I'd like to note that the empirical component of this paper is strong: I was impressed by the breadth of architectures and settings covered, and the experiments left me reasonably convinced that the classification layer can often be fixed, at least for image classification tasks, without significant loss of accuracy.

I have two general concerns. For one, removing the fully connected classification layer is not a novel idea; All Convolutional Networks (https://arxiv.org/abs/1412.6806) reported excellent results without an additional fully connected affine transform (just a global average pooling after the last convolutional layer). I think it would be worth at least referencing/discussing differences with this and other all-convolutional architectures. Including a fixed Hadamard matrix for the classification layer is I believe new (although related to an existing literature on using structured matrices in neural networks).

However, I have doubts about the ability of the approach to scale to problems with a larger number of classes, which arguably is a primary motivation of the paper ("parameters ... grow linearly with the number of classes"). Specifically, the idea of using a fixed N x C matrix with C orthogonal columns (such as Hadamard) is only possible when N > C. This is a critical point: in the N > C regime, a final hidden representation with N dimensions can be chosen to achieve *any* C-dimensional output, regardless of the projection matrix used (so long as it is full rank). This makes it seem fairly reasonable to me that the network can (at least approximately, and complicated by the ReLU nonlinearities) fold the "desired" classification layer into the previous layer, especially with a learned scaling and bias term. In fact it's not clear to me that the fixed classification layer accomplishes anything here, beyond projecting from N -> C (i.e., if N = C, I'd guess it could be removed entirely similar to all convolutional nets, as long as the learned scaling and bias were retained).

On the other hand, when C > N, it is not possible to have mutually orthogonal columns, and in general the output is constrained to lie in an N-dimensional subspace of the overall C-dimensional output space. Picking somewhat randomly a *fixed* N-dimensional subspace seems like a bad idea when N << C, since it is unlikely to select a subspace in which it is possible to adequately capture correlations between the different classes. This makes the proposed technique much less appealing for precisely the family of problems where it would be most effective in reducing compute/memory requirements. It also provides (in my view) a clearer explanation for the failure of the approach in the NLP setting. These issues were not discussed anywhere in the text as far as I can tell, and I think it's necessary to at least acknowledge that mutually orthogonal columns can't be chosen when C > N in section 2.2 (and probably include a longer discussion on the probable implications).

Overall, I think the paper provides a useful observation that clearly isn't common knowledge, since classification layers persist in many popular recent architectures. But the notion of fixing or removing the classification layer isn't particularly novel, and I don't believe the proposed technique would scale well to settings with many classes. As is I think the paper falls slightly short.

---

> ### Author Response · Authors · 2017-12-23
> **answer**
>
>
> We thank the reviewer for his detailed feedback on our paper.
> We hope to address the 2 main concerns raised:
> 1) Novelty - "removing the fully connected classification layer is not a novel idea; All Convolutional Networks (https://arxiv.org/abs/1412.6806) reported excellent results without an additional fully connected affine transform (just a global average pooling after the last convolutional layer)"
>
> We believe there is a slight misunderstanding here: in the "All convolutional networks" paper the fully-connected was not removed, as it just got replaced with a convolutional layer with the same number of parameters. This means there is still a final classifier (a conv layer) with number of parameters proportional to the number of classes.
> Our work introduces what we believe to be a novel idea - removing the classifier layer altogether making the number of network parameters independent from the number of classes. We added a clarification to this matter in our recent revision.
>
> 2) Applicability of our method when C > N:
>
> The reviewer is right in his claim that when C > N we can not have mutually orthogonal columns, but this is true even for a fully learned weight matrix.
> We empirically verified that for the vision use-cases brought in the paper we achieve good performance for C > N (e.g., on imagenet, so C=1000, with either mobilenet 0.5 where N = 512 or resnet with N reduced to 256).
> We do agree with the reviewer that this can be limiting when the classes have strong correlation with one another (as in the NLP case) and we add this as another possible explanation. We still, however, feel that this can be useful even for C >> N in other domains such as vision.

---

### Comment · AnonReviewer3 · 2017-11-20
**Quick comment**

One thing that is not clear to me from the paper are the experiments done with the Hadamard version AND scaling?

---

> ### Author Response · Authors · 2017-11-21
> **Answer**
>
> Yes, the experiments are done with both hadamard matrix and scaling.

---

### Public Comment · ~Gaurav_Sahu1 · 2018-11-30
**Doubt regarding updated loss function equation (Pg. 3)**

First of all, I thank the authors for such a nice paper. However, as I was going through it, I came across updated loss function equation:

L(x,t) = -alpha*q_t + b_t + logterm

where I feel that the sign for the bias term b_t should've been negative. Kindly correct me if I am wrong.

---

> ### Author Response · Authors · 2018-12-02
> **Thanks**
>
> Yes, you are correct, this is a typo. We will try to fix this.
>
> Thanks for noticing and letting us know.

---

> > ### Public Comment · ~Gaurav_Sahu1 · 2018-12-19
> > **Thank you**
> >
> > Thank you very much for addressing the doubt.

---

### Decision · Program_Chairs · 2018-01-29
**ICLR 2018 Conference Acceptance Decision**

**Decision:**

Accept (Poster)

**Comment:**

This paper proposes an interesting new idea which creates an interesting discussion.